# Task-Relevant Failure Detection for Trajectory Predictors in Autonomous Vehicles

**Alec Farid**[*,†,1], **Sushant Veer**[*,2], **Boris Ivanovic**[2], **Karen Leung**[2,3], **Marco Pavone**[2,4]

[1]Princeton University, [2]NVIDIA Research, [3]University of Washington, [4]Stanford University

afarid@princeton.edu, {sveer,bivanovic,kaleung,mpavone}@nvidia.com

**Abstract:** In modern autonomy stacks, prediction modules are paramount to planning motions in the presence of other mobile agents. However, failures in prediction modules can mislead the downstream planner into making unsafe decisions. Indeed, the high uncertainty inherent to the task of trajectory forecasting ensures that such mispredictions occur frequently. Motivated by the need to improve safety of autonomous vehicles without compromising on their performance, we develop a probabilistic run-time monitor that detects when a "harmful" prediction failure occurs, i.e., a *task-relevant* failure detector. We achieve this by propagating trajectory prediction errors to the planning cost to reason about their impact on the AV. Furthermore, our detector comes equipped with performance measures on the false-positive and the false-negative rate and allows for data-free calibration. In our experiments we compared our detector with various others and found that our detector has the highest area under the receiver operator characteristic curve.

**Keywords:** Run-time Monitoring, Autonomous Vehicles, Trajectory Prediction

## 1 Introduction

In general, autonomous vehicle (AV) stacks forecast the trajectories that non-ego agents—such as other vehicles, pedestrians, bicyclists, etc.—are likely to take. The accuracy of these predictions is crucial for the motion planner to ensure that the AV does not find itself in an unsafe state. Unfortunately, trajectory forecasting is an endeavor wrought with a high degree of uncertainty, a large portion of which, arguably, is aleatoric, arising from the stochasticity of human behavior. The situation is further exacerbated by the epistemic uncertainty that arises from unavoidable learning errors when training trajectory-prediction networks from finite driving logs. Therefore, it should come as no surprise that the trajectory-prediction networks frequently mispredict, leading to misguided motion plans. How then, in the face of such uncertainty, do we ensure safety of the AV without compromising its functionality? We approach this challenge by developing a *task-relevant* failure detector for the prediction module that only triggers if the estimate of the realized cost at a given time-instant is significantly worse than the cost predictions based on trajectory forecasting.

We require our detector to embody the following desiderata: First, the detector should be *task relevant* to reduce the number of unnecessary interventions that hurt the AV's functionality, i.e., the detector should only trigger if the failure is harmful to the AV's safety. To better illustrate task-relevance, consider the scenarios in Fig. 1. If the non-ego vehicle is predicted to stay in its lane, but it turns right, into the AV's lane (Fig. 1(a)), the cost for the planner would increase and therefore, the task-relevant detector should trigger. However, if the car turns left, away from the AV's lane (Fig. 1(b)), the cost for the planner would decrease, hence, the detector should not trigger. Note that in both the scenarios in Fig. 1, the predictions are incorrect, but they are task-relevant only in Fig. 1(a). Second, the detector should be an independent module, i.e., it should not modify the prediction module that it monitors. This enables easier integration of the detector with existing AV stacks and also facilitates the prediction module to specialize in its task without having to forego performance in favor of detectability of failures. Finally, the detector should be accompanied with a *metric to gauge its performance quality* to facilitate interpretable and trustworthy decision making. Most prior work satisfies only a subset of these desiderata [1, 2, 3, 4]. In this paper we propose an ap-

---

[*]Equal Contribution

[†]Work done as an intern with the NVIDIA Research Autonomous Vehicle Research Group

6th Conference on Robot Learning (CoRL 2022), Auckland, New Zealand.

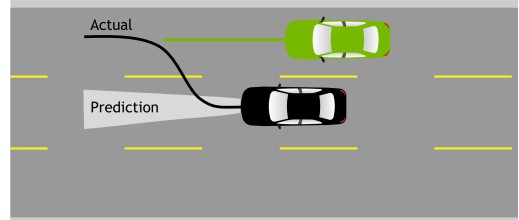 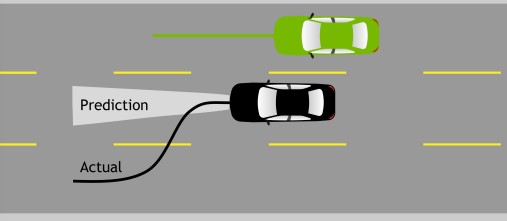

(a) Task-relevant prediction failure  (b) Non-task-relevant prediction failure

Figure 1: Illustration of task-relevant failure detection. The ego-vehicle and its motion plan is in green, the non-ego vehicle and the trajectory it takes is in black while the predicted trajectory distribution is in light-grey.

proach that satisfies all these three desiderata by reasoning about the effect of erroneous predictions further downstream at the planner module of the AV stack.

**Statement of Contributions.** The main contributions of this paper are as follows: (i) We formalize a *notion of anomaly* that underpins the design of a task-relevant prediction failure. (ii) We develop a *fast and efficient task-relevant failure detector* for the prediction module used in AV stacks which is simultaneously accompanied by *guarantees on the false-positive and false-negative detection rates* for improved interpretability. (iii) Owing to the interpretability of our approach, we can perform *data-free calibration* for our detector. Furthermore, we show in our experiments that our data-free calibrated detector's performance is not very far from the optimally calibrated detector using the Receiver Operator Characteristic (ROC) curve. (iv) We demonstrate the efficacy of our approach on nuScenes [5] and nuPlan teaser [6] datasets and provide comparisons with other detection methods in the literature. We find that our detector has the highest Area Under the ROC (AUROC) curve.

## 2   Related Work

**AV Safety.** Since autonomous vehicles require interactions with other road users and are safety-critical by nature, there has been a lot of effort, from regulators, industry, and researchers alike, in ensuring safe AV operations [7, 8, 9, 10, 11, 12, 13, 14, 15, 16] (see [17] for a review). A common approach is to compute "inevitable" collision sets (ICS), e.g., via Hamilton-Jacobi reachability computation [18], with selected assumptions on other agents' behaviors, and perform *shielding* where the AV will flag a situation as unsafe whenever it is close to entering the ICS and execute an appropriate evasive action (e.g., [1, 19, 20]). A primary challenge is selecting *reasonable* behavior assumptions for ICS computation to balance tractability, interpretability, and compatibility with real-world driving interactions [21, 22]. Most of these approaches assume that other agents act in an adversarial manner leading to overly conservative ICS. Instead, in this work, we build a run-time monitor that identifies only those prediction failures which are safety-critical in nature. Furthermore, the general detection approach we present could also be used as a run-time task-relevant safety monitor.

**Anomaly Detection.** Anomaly detection or change detection in signal processing literature has been an active area of work for decades (see [23] for a review). The goal is to identify events which deviate significantly from a reference input. Such methods have been applied in supervised learning settings to higher-dimensional inputs such as images [24, 25, 26] (see also [27] for a review); however, these methods are not task-relevant in nature. Other methods perform detection by estimating agent [28, 29] or input [30] atypicality. Recent techniques also allow for detection using the reward in reinforcement learning settings using streams of sensor inputs [4, 31, 32, 33, 34]. However, these methods do not provide guarantees on the false-positive and false-negative detection rates. Some methods that can provide guarantees come with caveats such as specific architectures [2, 35], only providing guarantees in retrospect (i.e. after multiple anomalies have occurred) [3], or on test data that is statistically similar to the training data [36]. We develop an *online* anomaly detector which provides guarantees for both false positive rate and false negative rate. Unlike many of the works described above, our detector does not require modifying the existing module on which it operates.

**Planning-aware Prediction.** The vast majority of behavior prediction works focus solely on the task of prediction (improving forecasting accuracy, incorporating additional sources of scene context, accelerating runtime, etc), in isolation from other components of the autonomy stack [37]. In recent years, however, there has been increasing interest in better integrating prediction with the rest of the robotic autonomy stack. Examples include works that more tightly integrate prediction with

object classification [38], tracking [39, 40], and planning [41, 42, 43, 44, 45, 46, 47]. While these approaches focus on better *architectural* integrations across the stack, a few recent works have also proposed to integrate modules *metrically*. For instance, [48, 49] present metrics that evaluate perception and prediction modules in a planning-aware fashion, e.g., by heavily weighting detection or prediction errors that could cause a collision. Our work is similar in that it can identify anomalous predictions which would detrimentally affect ego-vehicle planning, however, it does so online with guaranteed false-positive and false-negative detection rates.

## 3  Problem Definition

We express the dynamics of the AV and the scene around it as a Partially-Observable Markov Decision Process (POMDP) defined as a tuple $(\mathcal{X}, \mathcal{O}, \mathcal{U}, f)$ where $\mathcal{X}$ is the state space which is composed of the state of the ego agent $x^{\mathrm{e}}$, non-ego agents $x^{\mathrm{ne}}$, and other variables such as the environment map $x^{\mathrm{m}}$, $\mathcal{O}$ is the space of observations that the ego agent receives from its sensors, $\mathcal{U}$ is the space of the control inputs for the ego agent, and $f(x_t|x_{t-1}, u_{t-1})$ is the stochastic state-transition function.

Let $c_t : \mathcal{X} \times \mathcal{U} \to [0, \infty)$ be a cost function at a time instant $t$. Our framework is agnostic to the choice of the cost function; we only assume that the cost function is formulated such that higher values imply worse / less safe outcomes for the AV. As an example cost function is the closest distance of an AV to other non-ego agents and obstacles in the map. Given the predicted state $\hat{x}_{t+\tau}$ and the control input $u_{t+\tau}$ at $\tau$ time steps in the future, the predicted cost is $\hat{c}_{t+\tau} := c_{t+\tau}(\hat{x}_{t+\tau}, u_{t+\tau})$.

With a slight abuse of notation, let $x_t \in \mathcal{X}$ be an *estimate* of the state (instead of the true state) at time $t$ provided by the perception module by processing the sensor observation $o_t$. The prediction module takes in a history of state estimates $x_{0:t}$ and provides a distribution $\psi(\hat{x}^{\mathrm{ne}}_{t+1:t+T}|x_{0:t})$ on the predicted future state trajectories $\hat{x}^{\mathrm{ne}}_{t+1:t+T}$ of non-ego agents over the horizon $T$. The distribution produced by the prediction module induces a sequence of distributions $\{\phi_{t+\tau}(\hat{c}_{t+\tau}|x_{0:t})\}_{\tau=1}^{T}$ on the predicted cost $\hat{c}_{t+\tau}$ for each time-step in the prediction horizon $T$. The observed cost $c^{\star}_{t+\tau} := c_{t+\tau}(x_{t+\tau}, u_{t+\tau})$ at time-step $t + \tau$ is the cost estimated or "observed" by the AV given state-estimates of the ego and non-ego agents $x_{t+\tau}$ and the control input $u_{t+\tau}$. We define the notion of anomaly that will underpin our approach to prediction failure detection:

**Definition 1** ($p$-quantile anomaly). *For any time-step $t + \tau$, let $\phi_{t+\tau}(\hat{c}_{t+\tau}|x_{0:t})$ be the distribution on the predicted cost as described above. If the observed cost $c^{\star}_{t+\tau}$ lies in the top $p$-quantile of $\phi_{t+\tau}(\hat{c}_{t+\tau}|x_{0:t})$, then the prediction module suffers from a $p$-quantile anomaly.*

The observed cost $c^{\star}_{t+\tau}$ is, theoretically speaking, not necessarily drawn from the predicted cost distribution $\phi_{t+\tau}$ since the observed cost might depend on unmodeled/unobservable factors. However, under nominal circumstances (i.e., few or no task-relevant prediction errors and no major deviations by the AV from the open-loop motion plan), it is reasonable to expect the observed cost to be similar to the sampled predicted costs. This motivates the definition of anomaly as per Def. 1 and the design of a prediction failure detector based on the concept of $p$-quantile anomalies (illustrated in Fig. 2). Importantly, the notion of $p$-quantile anomaly, being a cost-based quantity, intrinsically captures the notion of task-relevance. Note that if the predictor predicts cost-degrading behavior correctly, the observed cost would not lie in top $p$-quantile as this is in alignment with the predictions.

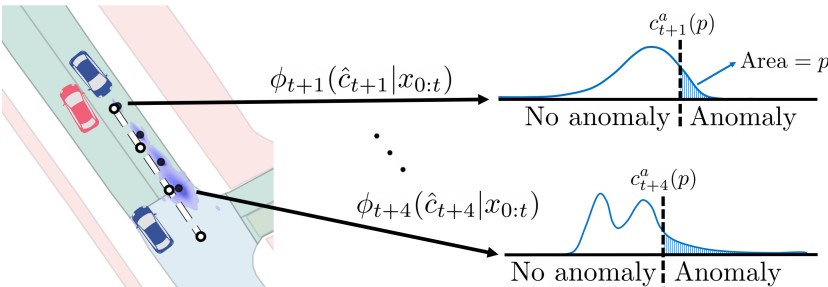

Figure 2: Illustration of the concept of $p$-quantile anomalies. The ego vehicle is red, non-ego vehicles are blue, ground truth trajectories are white, and trajectory predictions are the purple contour plots. At each time step, the cost function induces a distribution $\phi_{t+\tau}$ on predicted costs depicted by the blue plots on the right. If the observed cost $c^{\star}_{t+\tau}$ is within the top-$p$ quantile of $\phi_{t+\tau}$, it is a $p$-quantile anomaly.

# 4  $p$-**Quantile Anomaly Detector**

In this section, we present an algorithm for detecting $p$-quantile anomalies along with guaranteed false-positive and false-negative detection rates afforded by our algorithm. Let $c_{t+\tau}^{a}(p)$ be the cost which separates the top $p$ proportion of $\phi_{t+\tau}(\hat{c}_{t+\tau}|x_{0:t})$ from the bottom $1-p$. If $c_{t+\tau}^{\star} \geq c_{t+\tau}^{a}(p)$, then we have a $p$-quantile anomaly. The lack of an analytical expression[1] for $\phi$ prevents us from exactly computing $c_{t+\tau}^{a}(p)$, hence, we use a sampling-based approach for detecting if the observed cost lies within the $p$-quantile of the distribution on the predicted cost; see Fig. 2 for an illustration of the $p$-quantile anomaly within the trajectory prediction framework.

Suppose at time $t + \tau$ we sample $M$ predicted costs $S_{t+\tau} := \{\hat{c}_{t+\tau}^{1}, \ldots, \hat{c}_{t+\tau}^{M}\}$ independent and identically distributed (i.i.d.) from $\phi_{t+\tau}(\hat{c}_{t+\tau}|x_{0:t})$ and arrange them in an increasing order (i.e., $\hat{c}_{t+\tau}^{1} \leq \hat{c}_{t+\tau}^{2} \leq \cdots \leq \hat{c}_{t+\tau}^{M}$). Our detector $D$ returns a positive detection if the observed cost is greater than at least $M - n$ predicted costs in $S$, where $n \in \{0, \cdots, M - 1\}$:

$$D(n, S) := c_{t+\tau}^{\star} \geq \hat{c}_{t+\tau}^{M-n} \ . \tag{1}$$

Intuitively, by checking where the observed cost ranks against the sampled predicted costs, our detector estimates the quantile of $\phi$ in which the observed cost lies. In the rest of this section we present our detection algorithm, provide bounds on it's false-positive and false-negative detection rates, and use these bounds to analytically calibrate the detector by choosing appropriate $M$ and $n$.

## 4.1  Detection Algorithm

The detector, Algorithm 1, is executed at each planning cycle. We sample $M$ trajectories for each non-ego agent from the trajectory-prediction network $\psi$. Inference from prediction networks is usually a bottleneck in real-time control; however, note that for most AV stacks, we do not need to sample these trajectories specially for detection, as they would be drawn regardless for use in planning. Hence, by reusing these samples our detector does not generate significant computational overhead. At each planning cycle, our detector receives $M$ predicted costs for each non-ego agent at each prediction time-step (see line 1 in Algorithm 1). For each subsequent time-step in the prediction horizon, we monitor $D$ in (1). If the detection criteria is met, the detector returns `True` along with the identifier of the non-ego agent responsible for the detection. In Sec. 5.1, we show that this detector runs in less than $0.1$ milliseconds (ms) which is sufficiently fast for real-time monitoring.

---

**Algorithm 1** $p$-Quantile Anomaly Detector (QAD)

---

 1: **Input:** For each non-ego agent, a sequence of sets $S_{t+\tau} = \{\hat{c}_{t+\tau}^{1}, \cdots, \hat{c}_{t+\tau}^{M}\}$ for each $\tau \in \{1, \cdots, T\}$
 2: **Input:** Cost functions $c_{t+\tau}$ for each $\tau \in \{1, \cdots, T\}$
 3: **for** wall-clock time $w$ in $t + 1 : t + T$ **do**
 4:     **for** each non-ego agent **do**
 5:         $c_{w}^{\star} \leftarrow c_{w}(x_{w}, u_{w})$
 6:         **if** $c_{w}^{\star} \geq \hat{c}_{w}^{M-n}$ **then**
 7:             **return** (`True`, non-ego agent Id)
 8:         **end if**
 9:     **end for**
10: **end for**

---

## 4.2  Bounds on False-positive and False-negative Rates

Let us first define the false-positive rate (FPR) and the false-negative rate (FNR) for $D$.

**Definition 2** (False-positive Rate). *FPR of the detector is defined as the probability of drawing $S_{t+\tau}$ such that a $p$-quantile anomaly detection occurs when the anomaly does not occur, i.e.,*

$$FPR := \mathop{\mathbb{P}}_{S_{t+\tau} \sim \phi_{t+\tau}^{M}} [D(n, S_{t+\tau}) \wedge c_{t+\tau}^{\star} < c_{t+\tau}^{a}(p)] = \mathop{\mathbb{P}}_{S_{t+\tau} \sim \phi_{t+\tau}^{M}} [c_{t+\tau}^{\star} \geq \hat{c}_{t+\tau}^{M-n} \wedge c_{t+\tau}^{\star} < c_{t+\tau}^{a}(p)] \ . \tag{2}$$

---

[1] Even if the distribution on the predicted trajectories $\psi$ takes the form of a Gaussian mixture model [50], the nonlinear cost function may prevent the predicted cost distribution $\phi$ from acquiring an analytical form.

**Definition 3** (False-negative Rate). *FNR of the detector is defined as the probability of drawing $S_{t+\tau}$ such that a $p$-quantile anomaly detection does not occur when the anomaly does occur, i.e.,*

$$FNR := \mathop{\mathbb{P}}_{S_{t+\tau} \sim \phi_{t+\tau}^M} [\neg D(n, S_{t+\tau}) \wedge c_{t+\tau}^{\star} \geq c_{t+\tau}^{a}(p)] = \mathop{\mathbb{P}}_{S_{t+\tau} \sim \phi_{t+\tau}^M} [c_{t+\tau}^{\star} < \hat{c}_{t+\tau}^{M-n} \wedge c_{t+\tau}^{\star} \geq c_{t+\tau}^{a}(p)] . \quad (3)$$

Ideally, for safety-critical systems, such as AVs, we desire a low FNR to ensure that we do not miss any unsafe scenarios. This can be trivially achieved by having a detector that triggers all the time (high FPR), but clearly that is not desireable for the AV's performance. Hence, a good detector must also exhibit a low FPR. Motivated by this, we provide bounds on the FPR and FNR for our detector.

**Theorem 1** (FPR and FNR Bound). *Let the detector $D$ for the occurrence of a $p$-quantile anomaly be defined as in (1). Then, the FPR (Def. 2) and the FNR (Def. 3) for the detector satisfies:*

$$FPR \leq \sum_{i=0}^{n} \binom{M}{i} p^i (1-p)^{M-i} =: \overline{FPR} , \quad FNR \leq \sum_{i=n+1}^{M} \binom{M}{i} p^i (1-p)^{M-i} =: \overline{FNR} .$$

The proof of this theorem is presented in App. A. Thm. 1 ensures that if the detector identifies an anomaly (i.e., $D$ holds true), then with probability at least $1 - \overline{FPR}$, the observed cost does lie in the top-$p$ quantile of $\phi_{t+\tau}$, while if our detector does not detect an anomaly (i.e., $D$ is not true), then with probability at least $1 - \overline{FNR}$, the observed cost does not lie in the top-$p$ quantile of $\phi_{t+\tau}$.

Our detector satisfies all the three desiderata laid out in Sec. 1. First, our detector encodes the notion of *task-relevant failure detection* since it only triggers when the observed cost is high compared to the predicted costs, in the sense of the $p$-quantile anomaly (Def. 1). Second, the detector is *independent of the trajectory prediction module* that it monitors. Finally, we provide bounds on FPR and FNR of our detector in Thm. 1 that serve as a metric on the detector's performance. These bounds facilitate greater interpretability and promote easier detector calibration.

### 4.3 Data-free Detector Calibration

The analytical upper bounds for the FPR and the FNR in Thm. 1 allows data-free calibration of $D$ by letting us choose $M$ and $n$ so as allow a user to specify FPR and FNR values. However, we can only provide a strong bound on either the FPR or the FNR, but not both. This follows from the observation that the upper bounds in Thm. 1 satisfy $\overline{FPR} + \overline{FNR} = 1$. It is worth clarifying that this is *not* a fundamental limitation of our detector, but that of our analysis. We show in Sec. 5 that our detector, in fact, achieves low FPR and low FNR simultaneously.

## 5 Experimental Results

In this section, we benchmark the performance of our detector on nuScenes [5] and nuPlan teaser [6] datasets against various other methods. We study the detector in two applications: (i) run-time monitoring of task-relevant prediction network failures and (ii) filtering interesting scenarios from unlabelled expert driving logs. Through these experiments we demonstrate the ability of our detector to provide low FPR and low FNR for detecting task-relevant mispredictions. All experiments were conducted on a desktop computer with an Intel i9-10980XE CPU (18 cores) and 64 GB of RAM. Our code is available at: https://github.com/NVlabs/pred-fail-detector.

**Trajectory prediction module.** While our method is agnostic to the particular prediction method used, in our experiments we employ Trajectron++ [50], a state-of-the-art multi-agent trajectory forecasting model, to predict the motion of non-ego agents. It is a graph-structured recurrent neural network that predicts an agent's future position distribution given its past trajectory history and the past trajectories of its neighboring agents. At its core, Trajectron++ uses a conditional variational autoencoder (CVAE) to model the potential for multiple future trajectories.

**Detection and comparison methods.** We compare our detector—$p$-quantile anomaly detection (QAD)—against four other approaches: (i) *likelihood detection* baseline which triggers when the likelihood of the achieved states of an agent in the predicted GMM is below a threshold; (ii) *uniform and partial cost degradation* tests [4] which uses $p$-values to detect shifts in the cost distribution; (iii) *time-to-collision (TTC)* detector which triggers when the TTC drops below a threshold; and (iv) detection based on *Hamilton-Jacobi (HJ) reachability* analysis [1] which triggers if a collision is possible assuming an adversarial behavior by the non-ego agent. See Sec. B.1 for more information about the comparison methods.

## 5.1 Task-relevant Detection

**Overview.** We study the use of our detector and other baseline detectors mentioned above as run-time monitors for task-relevant prediction failure. The experiments are conducted in the recently released nuPlan teaser dataset [6] which allows the inclusion of reactive agents in the driving logs.

**Planner.** We assume a behavior generator provides us with a reference trajectory to follow. Our planner is then tasked with following this trajectory while avoiding collision with other agents on the road, ensuring ego comfort and satisfying the ego's dynamical constraints. The planner achieves this by generating a motion primitive tree [51] and choosing the trajectory that minimizes the weighted sum of a series of cost functions; further details on the planner are supplied in App. B.2. We use time-to-collision cost $c_{\text{ttc}}$ (computed assuming that agents continue moving at a constant speed in their current heading), momentum-shaped distance to other agents cost $c_{\text{d2a}}$, distance to goal cost $c_{\text{d2g}}$, distance to the reference trajectory cost $c_{\text{d2r}}$, speed limit cost $c_{\text{velocity}}$, ego comfort cost $c_{\text{comfort}}$, and reversing cost $c_{\text{reverse}}$ which penalizes the car for going against the heading direction of the lane:

$$c_{t+\tau} = w_1 c_{\text{ttc}} + w_2 c_{\text{d2a}} + w_3 c_{\text{d2g}} + w_4 c_{\text{d2r}} + w_5 c_{\text{velocity}} + w_6 c_{\text{comfort}} + w_7 c_{\text{reverse}} . \quad (4)$$

The exact expressions for the cost terms and the weights $w_1, \cdots, w_7$ used in the experiments are provided in App. B.3. The motion planner uses trajectory forecasts of other non-ego agents drawn from Trajectron++ to compute the momentum-shaped distance and the time-to-collision cost. The cost functions take about $1 \pm 0.5$ ms to evaluate. Note that these cost evaluations for planning are reused by our anomaly detector in Algorithm 1, thereby minimizing any excessive computational overhead by our detector. The anomaly detection computation takes less than 0.1 ms.

**Dataset and Labeling.** We test on 260 nuPlan [6] driving scenarios which comprises of a total of 2568 prediction horizons. We hand-label each prediction horizon for task-relevant prediction errors by visually inspecting plots of true trajectories of agents along with their predicted trajectories. We emphasize that the labeling is done merely by visual inspection and does not use the planning costs or prediction likelihoods. This is to ensure that the labeling procedure is independent of the detection methods to promote fair comparisons; see App. B.4 for more details on the labeling procedure.

**Remark 1** (Imbalance in Dataset). *Despite the use of reactive agents along with our custom planner in nuPlan, there is a strong bias in the dataset towards benign scenarios. Only 3.5% of this dataset is labeled as positive. Due to the imbalance, there is a bias toward detectors to exhibit lower FPR and higher FNR, since false-negative errors are amplified while the false-positive errors are mitigated.*

**Results.** We plot the Receiver Operator Characteristic (ROC) curve in Fig. 3 for each detector by varying: the detection rank $M - n$ in (1) for our detector, the $p$-value for UDT and PDT, the likelihood threshold for the likelihood-based detector, the TTC threshold for TTC detector, and the sub-level set of the value function for the HJ reachability based detector. The ROC curve for our detector is the closest to an ideal detector and has the highest Area Under the ROC (AUROC) curve.

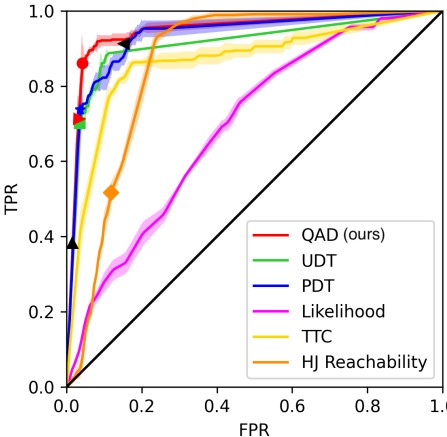

Figure 3: ROC curve for detectors. We report the mean of 5 trials, the shaded region is 2 times the standard deviation.

Table 1: AUROC and best point from ROC (↑ AUROC)

| Detection Method | AUROC | FPR % | FNR % |
|---|---|---|---|
| QAD (ours) | **0.946±0.002** | **8.4±0.4** | 7.9±0.7 |
| PDT [4] | 0.932±0.005 | 18.5±0.2 | **5.8±1.1** |
| UDT [4] | 0.912±0.002 | 10.6±0.3 | 11.5±0.4 |
| TTC | 0.864±0.004 | 17.3±0.5 | 13.7±0.4 |
| HJ Reachability [1] | 0.862±0.003 | 23.7±0.3 | 7.2±0.6 |
| Likelihood | 0.693±0.006 | 45.9±0.4 | 24.5±0.4 |

Table 2: Data-free calibration (↑ distance from $y = x$)

| Detection Method | | FPR % | FNR % | Dist. |
|---|---|---|---|---|
| QAD (FNR) (ours) | ● | 4.2±0.3 | 13.9±1.1 | **0.58±0.01** |
| FNR Ensemble (ours) | ◀ | 15.2±0.3 | **8.7±1.2** | 0.54±0.01 |
| PDT (FPR) [4] | ★ | 3.7±0.3 | 26.3±0.6 | 0.50±0.01 |
| QAD (FPR) (ours) | ▶ | 3.4±0.3 | 28.8±0.9 | 0.48±0.01 |
| UDT (FPR) [4] | ■ | 3.5±0.3 | 29.9±0.9 | 0.47±0.01 |
| HJ Reachability [1] | ◆ | 11.8±0.2 | 48.6±1.7 | 0.28±0.01 |
| FPR Ensemble (ours) | ▲ | **1.5±0.3** | 61.6±1.8 | 0.26±0.01 |

We obtain the best calibration for each detector by choosing the point that is farthest from the $y = x$ line in Fig. 3 and report the empirical FPR and FNR for them in Table 1. Our detector has the lowest FPR and the third-lowest FNR. PDT has a slightly better FNR than our method, but the FPR is much higher, while for all other methods both FPR and FNR are higher than our method. As one would expect, the naive likelihood baseline does not perform very well while the HJ reachability detector is overly conservative exhibiting a lot of false-positives.

We also compare the performance of our detector with data-free calibration using Thm. 1 with the data-free calibration of UDT, PDT, and HJ reachability. For our detector, we consider an anomaly to have occurred when the true cost lies in the top-5% of the predicted cost distribution (see Def. 1); we chose the 5% quantile because it is a commonly used threshold for detection in statistics. We include two versions of our detector in this study: (i) calibrated to bound the FPR below 5% and (ii) calibrated to bound the FNR below 5% using Thm. 1; as discussed in Sec. 4.3, we cannot simultaneously obtain a strong bound on both FPR and FNR. We use a $p$-value of 0.05 for UDT and PDT, and a 0-sublevel set of the value function for HJ reachability. Additionally, we construct an FPR ensemble detector, which only detects positive if *all* of QAD (FPR), UDT, PDT, and HJ reachability are positive, and an FNR ensemble detector, which detects positive if *any* of QAD (FNR), UDT, PDT, and HJ reachability are positive. For the ensemble detectors, we use the data-free calibration for each method described above.

These detectors are labeled using markers in the ROC curves in Fig. 3. Their empirical FPR and FNR is reported in Table 2 and we sort the table by distance from the line $y = x$. We observe that our FPR bounded detector satisfies the 5% bound provided by Thm. 1; however, our FNR bounded detector violates the bound. This disagreement between the theory and the empirical performance for FNR occurs because of the mismatch between our anomaly criteria of top-5% quantile and what the human data labeller considers to be an anomaly. A higher empirical FNR suggests that the anomalous quantile should be expanded beyond 5%; case in point, the optimal $p$-quantile anomaly detector obtained using the ROC curve in Table 1 with 7% FNR corresponds to approximately the top-35% quantile as an anomaly. Furthermore, the imbalance in the dataset also contributes to this problem, as discussed in Rem. 1. More data as well as a more balanced dataset can perhaps remedy this mismatch and lower the FNR; we will explore this in future work. Despite this mismatch, we see in Table 2 that our FNR detector performs the best out of all detectors in terms of distance to the $y = x$ line in the ROC plot. The FNR ensemble and PDT also perform well, favoring false-negatives and false-positive respectively. Since false-negatives can be safety critical, a designer may choose our FNR ensemble detector at the cost of more false-positives. Ultimately, our FNR detector is not very far off from the optimal detector obtained from the ROC curve, i.e., with *no data at all* we are able to calibrate our detector using Thm. 1 to perform very well.

## 5.2 Filtering expert driving logs

**Overview.** Given driving data logs, we consider the task of filtering scenarios with anomalies that may be of interest for robsutifying the prediction network or for diagnostic evaluations. We use the nuScenes and nuPlan datasets as the driving logs. For the likelihood detector we choose the threshold 0.05, and for TTC we choose 1 second, while for all other detectors, we use the data-free calibration specified in Sec. 5.1.

**Dataset.** The dataset consists of 375 nuPlan and 152 nuScenes scenarios with a total of 3890 prediction horizons, in all of which the ego-vehicle is driven by a human driver and there are no collisions.

**Cost functions.** Since the nuScenes and nuPlan driving logs are collected by a human driver, we do not have access to the true planning "cost" used by the driver. Instead, we use a weighted sum of cost terms contributed by the momentum-shaped distance and the time-to-collision for each non-ego agent as a proxy cost: $c_{t+\tau} = w_1 c_{ttc} + w_2 c_{d2a}$. The exact expressions of $c_{ttc}$ and $c_{d2a}$ are provided in App. B.3. Our results suggest that this cost function is well-aligned with human intuition as it allows us to filter out interesting scenarios while ignoring the uninteresting ones.

**Results.** We compare our detector to all baseline methods described earlier. See Table 3 for a comparison of the methods and Fig. 4 for representative examples of positive detections. We observe that the likelihood detection and the TTC-based detection both have many more positives than the other methods, followed by HJ reachability. Other methods have a similar number of positives. Ultimately, no driving scenario from this dataset is unsafe, so we inspect the positive detections

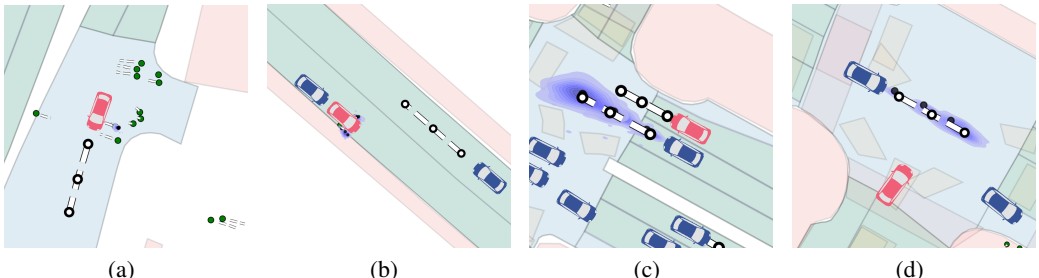

Figure 4: Selected scenarios from nuScenes. The ego vehicle is red, non-ego vehicles are blue, pedestrians are green circles, ground truth trajectories are white, and Trajectron++ predictions are the purple contour plots.

from each method to find interesting scenarios. All methods detect a handful of interesting scenarios, such as Fig 4(b) where pedestrians are right beside the ego (possibly entering or exiting). We also see additional interesting scenarios detected by our method but not HJ reachability such as in Fig. 4(a) where a close-by pedestrian is predicted to stand still, but moves quickly towards the ego at a crosswalk. Qualitatively, it seems that our detector, UDT, and PDT all do a good job of identifying interesting scenarios such as those shown in Fig. 4(a) and (b). We see that HJ reachability, and likelihood detection methods more frequently labels driving scenarios where the predictions are accurate and there is nothing inherently unsafe as anomalous such as in Fig. 4(c) and (d).

Table 3: Detections from expert driving logs

| Detection Method | Percentage Detected |
|---|---|
| UDT [4] | $5.4 \pm 0.1$ |
| QAD (FPR) (ours) | $5.4 \pm 0.1$ |
| PDT [4] | $5.8 \pm 0.1$ |
| QAD (FNR) (ours) | $6.4 \pm 0.1$ |
| HJ Reachability [1] | $8.8 \pm 0.0$ |
| TTC | $17.4 \pm 0.0$ |
| Likelihood | $58.5 \pm 0.0$ |

## 6   Limitations, Future Work, and Conclusion

**Limitations.** As discussed in Sec. 4.3, we cannot provide a strong FPR and FNR guarantee simultaneously for our detector. This does not seem to be a fundamental limitation of the detector itself because our empirical results in Table 1 suggest that we can achieve low FPR and low FNR. As part of our future work, we intend to explore providing stronger guarantees on both FPR and FNR simultaneously. A second limitation of our approach is identifying what quantile of the cost distribution corresponds to an actual anomaly. As we saw in Sec. 5.1, the mismatch in the quantile can result in a disagreement with the theoretical bounds. From a practical perspective, this is not a major limitation as we demonstrated that our detector performs well despite making a completely uninformed quantile choice for anomaly detection. Furthermore, we can mitigate this problem by using a small dataset to identify an appropriate $p$-quantile to be considered an anomaly.

**Future Work.** As part of our future work, we hope to use this detector for adapting planning frequencies. If the planner re-plans only when necessary, it would free up some of the compute budget for other modules in the AV stack and lower the energy consumption. Since our detector directly monitors the cost of the planner, it would serve as a good indicator of when the motion plan is not evolving as expected and therefore, requires re-planning. As a proof of concept, we present some motivating results in App. B.6. An additional direction for future work is to require multiple $p$-quantile anomalies before declaring a detection instead of a single one used in this work. Another exciting direction for this work is providing causal explanations for the detections; e.g., we anticipated the agent to continue straight, but it is switching into our lane. Such explanations can actively be leveraged to plan reactions to prediction failures.

**Conclusion.** We presented a task-relevant detector for mispredictions by the trajectory forecasting networks used in AVs. Our detector builds a distribution of the predicted future costs and identifies an anomaly when the true cost of the planner lies in the top-p quantile of that distribution. We provided FPR and FNR guarantees for our detector that facilitate data-free calibration. In experiments, we demonstrated that our detector out-performs all other baseline approaches we compared against and is compute-efficient. Additionally, we showed that our data-free calibrated detector's performance is close to the optimal detector chosen from the ROC curve. We used our detector for run-time misprediction monitoring and for filtering prediction failures from unlabeled driving logs.

**Acknowledgments**

The authors are grateful to Anirudha Majumdar for helpful feedback and discussions on this work.

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
