# OpenReview forum: "Task-Relevant Failure Detection for Trajectory Predictors in Autonomous Vehicles"
_robot-learning.org/CoRL/2022/Conference — CoRL 2022 Poster_

### Official Review · Reviewer_nVyF · 2022-07-07

**Originality:** Good
**Technical Quality:** Good
**Clarity Of Presentation:** Good
**Impact:** 3

**Recommendation:**

Weak Accept: I recommend accepting the paper, but will not argue for my recommendation if the majority of other reviewers have a different opinion.

**Summary:**

In the paper a task-relevant failure detector for monitoring trajectory prediction is proposed. The novelty is, that prediction errors are detected by evaluating the effect of the prediction on the subsequent planning task. A trajectory prediction is ranked abnormal if the planning cost is larger than some threshold. More precise, given the distribution of the prediction module, they propose a sampling based evaluation of the cost distribution to define the p-quantile cost threshold. If the estimated cost given the control inputs of the planning module exceeds the p-quantile threshold, the detector returns a positive detection, i.e. predicts an abnormal event.

They provide theoretical bounds for the false-positive rate (FPR) and false-negative rates (FNR) of the detector which can be used for data-free calibration of the detector, given either FPR or FNR.

**Issues:**

I have only minor issues (beyond my suggestions in "Strength and Weaknesses") that I would like the authors to address during revision

- It would be helpful if the notion of the cost terms in eq. (4) is included in the text L.209 - L.212.
- App. B2 L.504: error in term $15^{(T-1)}$
- To favor open research, it would be highly appreciated if code and dataset will be made public.

**Quality Of The Limitations Section:**

Limitations are addressed clearly

**Reviewer Expertise:**

4: The reviewer is confident but not absolutely certain that the evaluation is correct

**Robotics Focus:**

Highly relevant to robotics but no hardware experiments

**Strengths And Weaknesses:**

### Strengths
In my opinion the paper makes a solid contribution both technically and empirically, while the technical contribution is major. Some comments on the strength:

- Well written and well-defined problem definition.
- Considering anomaly detection from a task-relevance perspective is a reasonable approach. The influence of errors in perception and prediction on the actual driving task is of major interest. For example, a wrong prediction of a vehicle in far distance of the ego is counted as prediction failure under many trajectory prediction protocols, however might not be a task-relevant failure, if it doesn't hurt the safety of the AV.
- In AV safety, methods with theoretical proofs are highly valuable. The method allows to compute bounds on the FPR and FNR, although theoretical guarantees cannot be provided simultaneously on both quantities, but either on FPR or FNR.  The analysis of the limitations of the guarantees is noteworthy.

### Weaknesses
I have some suggestions for improving the paper:

- The paper addresses the detection of trajectory prediction failures which lead to high costs in the planning module. I wonder how failures in prediction are distinguished from real driving anomalies, for example the predictor predicts a sharp manouver because a driver shows rude or inattentive behavior (which is not a prediction error but still results in high planning cost). A discussion would be useful.
- The approach is compared to multiple baselines for detecting the anomaly: (1) Evaluating the *likelihood* of an achieved agent state under the GMM distribution of Trajectron++, (2) detecting shifts in the cost distribution using *uniform and partial cost degradation*, (3) detecting anomalies with low *time to collision (TTC)* and (4) *Hamilton-Jacobi reachability*. Recently, deep learning methods dominate anomaly detection, in particular trained methods to detect anomalies in multi-agent trajectories [1, 2]. For a complete picture, the comparison to one of the methods would be very useful.
- A qualitative evaluation of the failure detector on the nuPlan teaser dataset is missing. It would enhance the paper if success and failure cases of the detector would be shown.

[1] Wiederer, Julian, et al. "Anomaly Detection in Multi-Agent Trajectories for Automated Driving." Conference on Robot Learning. PMLR, 2022.

[2] Bera, Aniket, Sujeong Kim, and Dinesh Manocha. "Realtime anomaly detection using trajectory-level crowd behavior learning." Proceedings of the IEEE conference on computer vision and pattern recognition workshops. 2016.

**Summary Of Recommendation:**

The paper is well written and the proposed approach is both interesting and applicable to any trajectory prediction module. The support of the empirical evaluation could be improved.

---

> ### Author Response · Authors · 2022-08-22
> **Reply to Reviewer nVyF**
>
> Thank you very much for your valuable feedback. We address each of your points below.
>
> “The paper addresses the detection of trajectory prediction failures which lead to high costs in the planning module. I wonder how failures in prediction are distinguished from real driving anomalies, for example the predictor predicts a sharp maneuver because a driver shows rude or inattentive behavior (which is not a prediction error but still results in high planning cost). A discussion would be useful.”
> - This is a good point and worth further discussion. The main objective of this detector is to weed out faulty predictions that are potentially harmful to the AV. If the predictor predicts erratic behavior correctly, the observed cost would not lie in the p-tail of the predicted cost distribution as this is in alignment with the predictions. Therefore, even though this erratic behavior by another driver might result in higher overall costs, the detector would work as desired and not trigger (since there is no prediction error). We have added this note to the discussion below Def. 1 to improve clarity (we provide an updated paper in the comment titled “Updated Manuscript”).
>
> “Recently, deep learning methods dominate anomaly detection, in particular trained methods to detect anomalies in multi-agent trajectories [1, 2]. For a complete picture, the comparison to one of the methods would be very useful.”
> - Thank you for pointing out these papers, we have updated the related work to include them as they are very relevant. However, [1] generates a set of normal and abnormal maneuvers in order to train their anomaly detector. We do not assume access to any abnormal driving behavior for our methods. Furthermore, their definition of anomaly is not necessarily task-relevant. The objective of [2] is very different from that of our paper. They are trying to identify the most anomalous agent in a scene with multiple agents, while our paper identifies when the prediction for an agent is incorrect. To clarify the difference between the two objectives further, consider a scenario with just a single non-ego agent – the methods from [2] cannot detect the lone non-ego agent as anomalous, even though the predictions of that non-ego agent might be incorrect and harmful to the AV. Hence, [2] cannot be used for detecting prediction failures.
>
> “A qualitative evaluation of the failure detector on the nuPlan teaser dataset is missing. It would enhance the paper if success and failure cases of the detector would be shown.”
> - We appreciate this suggestion and have included Figure 6 in Appendix B which includes representative false positives, false negatives, true positives, and true negatives from the nuPlan teaser dataset in order to provide a better qualitative evaluation of our detector.
>
> Minor issues
> - Thank you for providing this additional feedback and for catching typos in the draft. We have added notation in L210-L213 (and leave additional details on the cost functions to Appendix B.3) and have fixed the typo you pointed out. We will make the code publicly available if the paper is accepted for publication.
>
> Overall, we reiterate that our detector improves the safety of the AV stack by identifying potentially safety-critical prediction errors. We also want to emphasize that in addition to the theoretical and experimental contributions, since our method is computationally lightweight, modular, and prediction model agnostic, it is easy to integrate in any AV stack that leverages multi-agent behavior prediction. Hence, we expect this work to have significant adoption in AV stacks.

---

> > ### Comment · Reviewer_nVyF · 2022-08-23
> > **Reply to Paper417 Authors**
> >
> > Thank you very much for your comprehensive reply to my review. In my opinion the additional qualitative results and discussions increase clarity. I appreciate the code will be made available to the public upon acceptance.
> >
> > Just a few comments:
> >
> > *"If the predictor predicts erratic behavior correctly, the observed cost would not lie in the p-tail of the predicted cost distribution as this is in alignment with the predictions."*
> >
> > - Thank you for explaining the behavior of your method in cases of correct predictions in irregular driving scenarios. This is really helpful.
> >
> >
> > *"However, [1] generates a set of normal and abnormal maneuvers in order to train their anomaly detector. We do not assume access to any abnormal driving behavior for our methods."*
> >
> > - You mention the paper uses both normal and abnormal maneuvers for training the anomaly detector. As far as I understood, the method proposed in [1] is trained in an unsupervised manner, such that they are not using any labels of abnormal driving during training and only train on normal driving scenarios. Abnormal maneuvers are generated to quantify the ability of detecting anomalies during testing. Of course, different from your porposed work, the definition of an anomaly in [1] is not necessarily task relevant.

---

> > > ### Author Response · Authors · 2022-08-23
> > > **Reply to Reviewer nVyF**
> > >
> > > We are glad to hear that our revision and responses were positively received.
> > >
> > > Thank you very much for clarifying [1]! Indeed, as the reviewer suggests, the notion of anomaly in [1] is significantly different from ours. In particular, [1] focuses on identifying an anomalous scene (or agent in a scene) while our detector focuses on identifying trajectory prediction errors. To detect trajectory prediction errors, we would have to perform detection in the trajectory space after making predictions, as opposed to detection in the latent space prior to predictions, as was done in [1]. This would involve a non-trivial extension.

---

### Official Review · Reviewer_dWwY · 2022-07-25

**Originality:** Fair
**Technical Quality:** Fair
**Clarity Of Presentation:** Very Good
**Impact:** 2

**Recommendation:**

Weak Accept: I recommend accepting the paper, but will not argue for my recommendation if the majority of other reviewers have a different opinion.

**Summary:**

1. The authors develop a probabilistic run-time failure detector for the prediction module of autonomous driving.
2. The main characteristic of this detector is "task-relevant". Namely, by incorporating the planning cost in the detector, the detector is designed to detect false predictions leading to the planning cost deviation.
3. The authors provide theoretical proof for the FPR, FNR bounds of the proposed detector.
4. The proposed detector is validated in two datasets and compared to several baseline methods.

**Issues:**

1. Justify whether the detector is sensitive to the cost function design. Try different cost function (with different terms and weights) and show whether a global M & n setup will work.

2. Justify how to choose M? Try complicated scenarios (multiple non-ego agents at intersection) & visualize the predicted cost distribution estimated from M predicted trajectories.

3. How the probability of the predicted trajectory will be incorporated?

4. Evaluate the consequence of an FN or FP event. What does it mean by 8% FP or FN rates? Should we abandon the prediction results when a failure is detected (or at what FP level we should abandon the prediction results)?

5. Justify the delay of the detection?  **What is the average delay for the detector** (for different M & n & FPR & FNR) ? What if there is only one frame of abnormal prediction and then the prediction is automatically recovered?  When we find one positive, should we regard it as a failure? (when the detector finds there is a failure, the prediction may be already normal).


**Quality Of The Limitations Section:**

Additional details required

**Reviewer Expertise:**

5: The reviewer is absolutely certain that the evaluation is correct and very familiar with the relevant literature

**Robotics Focus:**

Sufficient demonstration on hardware

**Strengths And Weaknesses:**

### Strengths

1. The idea of incorporating planning costs to measure task relevance is good.

2. The problem of detecting prediction failures is essential.

3. The paper is easy to understand and well written.

### Weaknesses

1. Although it claimed that "our framework is agnostic to the choice of the cost function", the detector may be **sensitive to the cost function design**. Eq. (1) essentially evaluates how the estimated cost (cost evaluated from state estimated) is ranked among the predicted costs (costs evaluated from the M sampled predicted trajectories). M & n are the hyperparameters for this detector. If the cost function is tuned a little bit, the predicted cost distribution may be completely different.  Take the cut-in case in Fig.1 (a) for example, the planning cost can be either tuned to have a large safety margin or a small one (actually this value may be  adjusted by the user on the fly), to obtain a reasonable balance between FPR and FNR, the M & n may need to be tuned. Should we tune individual detectors for every cost function? ( it may not be realistic). Or one universal setup of M & n will work for all cost function designs?  (the answer may be no since the cost distribution is different and something should be tuned for better FNR and FPR balance) *This may be an issue preventing the detector from working in a real autonomous driving system since the cost function is complicated and tuned very frequently.*

2. For complicated driving scenarios, **how to choose M?** Should we fix the M for all scenarios? For example, for a number of non-ego interactive agents in an intersection (every agent may have multimodal predictions), a very large M may be needed to yield a good estimation of the predicted costs. Will a very large M also work for normal on-lane driving?

3. How the probability of the predicted trajectory will be incorporated?

4. The metrics for evaluation are not strong. FNR/FPR/ROC only represents the precision-recall & accuracy of the detector.  What is the consequence of an FN or FP event? What does it mean by 8% FP or FN rates? Should we abandon the prediction results when a failure is detected (or at what FP level we should abandon the prediction results)?

5.  The failure detection is done by post-validating predicted planning costs against true costs. The detection is always delayed compared to the first frame of abnormal prediction. **What is the average delay for the detector** (for different M & n & FPR & FNR) ? What if there is only one frame of abnormal prediction and then the prediction is automatically recovered?  When we find one positive, should we regard it as a failure? (when the detector finds there is a failure, the prediction may be already normal).


**Summary Of Recommendation:**

1. The idea of task-relevant failure detection is good.
2. The current design of the detector is too simple to work in real autonomous driving vehicles, as listed in the weaknesses.
3. More thorough evaluation should be presented.
4. Still a gap to a solid contribution.

---

> ### Author Response · Authors · 2022-08-22
> **Reply to Reviewer dWwY**
>
> Thank you very much for your valuable feedback. We address each of your points below in two responses.
>
> “The detector may be sensitive to the cost function design… If the cost function is tuned a little bit, the predicted cost distribution may be completely different. Take the cut-in case in Fig.1 (a) for example, the planning cost can be either tuned to have a large safety margin or a small one (actually this value may be adjusted by the user on the fly), to obtain a reasonable balance between FPR and FNR, the M & n may need to be tuned. Should we tune individual detectors for every cost function? ( it may not be realistic). Or one universal setup of M & n will work for all cost function designs? (the answer may be no since the cost distribution is different and something should be tuned for better FNR and FPR balance) This may be an issue preventing the detector from working in a real autonomous driving system since the cost function is complicated and tuned very frequently.”
> - Indeed, a universal setup of M and n should work for most cost functions and this is one of the main strengths of our approach. Towards this end, Theorem 1 provides a data-free calibration for our method which only depends on the desired FPR / FNR bound and the quantile of the distribution to be considered an anomaly. This calibration does not need to be tailored to the cost function or encountered scenarios. Regardless of the cost function used, if the observed cost lies in the top-p quantile of the predicted cost distribution, this would indicate a p-quantile anomaly. Furthermore, we would like to highlight that the challenge involved in tuning hyperparameters to specific cost functions and scenarios was a primary motivation behind the development of the p-quantile anomaly detection approach presented in this paper.
>
> “For complicated driving scenarios, how to choose M? Should we fix the M for all scenarios? For example, for a number of non-ego -interactive agents in an intersection (every agent may have multimodal predictions), a very large M may be needed to yield a good estimation of the predicted costs. Will a very large M also work for normal on-lane driving?”
> - As mentioned above, we pick a single M (and n) for all scenarios using Theorem 1 only based on the desired bounds on the FPR / FNR. A larger M allows for a smaller bound on the FPR / FNR and is therefore preferable. Additionally, since we check for anomalies in each agent (see Algorithm 1 in the paper), a fixed M can be used (and was in the paper) for scenarios with any number of agents. Finally, we note that our results use all scenarios from nuScenes (some of which have 80+ agents). The number of agents is not a limitation on the detector.
>
> “How the probability of the predicted trajectory will be incorporated?”
> - We sample predictions independently and identically distributed (i.i.d.) in order to provide guarantees on the FPR and FNR of our detector. Therefore probabilities of predictions are incorporated automatically as higher probability trajectories are sampled more frequently than lower probability trajectories.
>
> “The metrics for evaluation are not strong. FNR/FPR/ROC only represents the precision-recall & accuracy of the detector.”
> - FPR, FNR, and ROC are standard metrics in the literature for detector evaluation (see [2,3,4] in the paper for example) and are, in general, considered sufficient for assessing the quality of  a detector. The practical significance of these metrics is as follows: the FPR provides the probability that the detector predicted a failure where there isn’t one, the FNR provides the probability that the detector did not predict a failure when there is one, and the ROC curve gives additional insight about the tunability of the detector thresholds and its impact on FPR and FNR. However, if the reviewer has suggestions for any additional metrics that can provide further insight, we would be very happy to try them out.
>
> “What is the consequence of an FN or FP event? What does it mean by 8% FP or FN rates?”
> - A FN event occurs when there is a failure but the detector fails to identify it. A FN event, if safety-critical in nature, can pose a safety risk for the AV. A FP event occurs when there is no failure, but the detector thinks there is. A FP event is detrimental to the performance of the AV since a detector with a high FP rate will frequently trigger interventions in the AV stack’s operation. Therefore, we desire a low FPR and a low FNR. An 8% FPR or FNR means that there is an 8% probability that the detector’s output is a FP event (if there is an affirmative detection) or a FN event (if there is a missed detection).

---

> > ### Author Response · Authors · 2022-08-22
> > **Reply to Reviewer dWwY continued**
> >
> > "Should we abandon the prediction results when a failure is detected (or at what FP level we should abandon the prediction results)?”
> > - We note that we use the FP and FN rates to quantify the detector and guarantees offline, they are not used online since they are aggregate metrics. If an anomaly is detected online, we could re-plan, fallback to a simple backup predictor (e.g., constant velocity), or trigger a safety-preserving maneuver. In this paper, we do not use the anomaly signal to change the planner, but this is an exciting direction for future work. We discuss the potential of using the anomaly signal provided by our approach for planning in the future work in Section 6.
> >
> > “The failure detection is done by post-validating predicted planning costs against true costs. The detection is always delayed compared to the first frame of abnormal prediction. What is the average delay for the detector (for different M & n & FPR & FNR)?”
> > - We mention in Line 219 that after the achieved cost is observed, the delay in detection is less than 0.1 milliseconds (ms) for M = 100, which is significantly faster than the perception, prediction, and control loops. This is the time it takes for the detection after the first frame of prediction error. This time does not depend on FPR or FNR since they are not computed online and are only used as metrics to evaluate our detector offline. As the predicted costs are computed while planning and re-used for detection, after the achieved cost is measured, we only need to compute the rank of the achieved cost in the predicted costs---this operation is performed very efficiently with a vectorized comparison (of size M).
> >
> > "What if there is only one frame of abnormal prediction and then the prediction is automatically recovered? When we find one positive, should we regard it as a failure? (when the detector finds there is a failure, the prediction may be already normal)."
> > - We thank the reviewer for bringing up this great point. However, choosing an ideal window for anomaly detection in a signal is an open problem which is beyond the scope of this paper. Currently, our detector would trigger even if it is for a single frame. Although this is a conservative approach, in practice we still see good performance, as evidenced by the low FPR (i.e., low rate of detecting an anomaly where there isn’t one). We have added this idea to the discussion in the future work in Section 6 (we provide an updated paper in the comment titled “Updated Manuscript”).
> >
> > “The current design of the detector is too simple to work in real autonomous driving vehicles”
> > - Our method is simple yet effective, as demonstrated in our many results and comparisons. In fact, we see the simplicity of our approach as a major strength. AV stacks have stringent compute constraints, hence the light-weight and simple nature of our detector facilitates easy integration in the AV stack.
> >
> > Overall, we reiterate that our detector improves the safety of the AV stack by identifying potentially safety-critical prediction errors. We also want to emphasize that in addition to the theoretical and experimental contributions, since our method is computationally lightweight, modular, and prediction model agnostic, it is easy to integrate in any AV stack that leverages multi-agent behavior prediction. Hence, we expect this work to have significant adoption in AV stacks.

---

> > > ### Author Response · Authors · 2022-08-26
> > > **Follow-up Response**
> > >
> > > Dear Reviewer,
> > >
> > > Thank you very much for your valuable feedback! Kindly let us know if we have addressed all your concerns. We would be happy to discuss any unresolved points.

---

> > > > ### Comment · Reviewer_dWwY · 2022-08-27
> > > > **Response to Authors**
> > > >
> > > > Dear Authors,
> > > >
> > > > Thanks a lot for your efforts and detailed explainations. Lots of points are clarified.

---

### Official Review · Reviewer_t48G · 2022-07-31

**Originality:** Good
**Technical Quality:** Good
**Clarity Of Presentation:** Very Good
**Impact:** 2

**Recommendation:**

Weak Accept: I recommend accepting the paper, but will not argue for my recommendation if the majority of other reviewers have a different opinion.

**Summary:**

This paper studies the problem of anomaly detection for trajectory predictors in autonomous vehicles. A probabilistic framework that detects task relevant failures, by propagating trajectory prediction errors to the planning cost is developed. To be able to distinguish between harmful and non-harmful mispredictions, the authors design of a prediction failure detector based on the concept of p-quantile anomalies. The proposed detector comes equipped with performance measures on the false-positive and the false-negative rate and allows for data-free calibration. To benchmark the performance of the anomaly detector, the authors compared their approach against various other methods on nuScenes and nuPlan datasets .The authors demonstrated that the proposed detector outperforms all other baseline approaches and is compute-efficient.

**Issues:**

Listed above as suggestions for improvement.



**Quality Of The Limitations Section:**

Additional details required

**Reviewer Expertise:**

3: The reviewer is fairly confident that the evaluation is correct

**Robotics Focus:**

Relevant but unlikely to deploy to hardware in near future

**Strengths And Weaknesses:**

Identified strengths of the paper:
     - Design of a prediction failure detector based on the concept of p-quantile anomalies.
     - FPR and FNR to guarantee that the anomaly detector facilitates the data-free calibration.
     - The obtained results can, in principle, be reproduced. Even though key resources (e.g., proofs, code, data) are unavailable.

Weakness/ Suggested imporvement of the paper:
      - The authors use Trajectron++, which is based on a conditional variational autoencoder (CVAE) to model the potential for multiple
         future trajectories. However, the sampling is based solely on likelihood which may only produce samples that correspond to the
         major modes of the data distribution. How can you garantee the diversity of the set of samples ?
      - For the results in Tab.3 it would be useful to provide a measure of statistical dispersion (e.g. standard deviation or standard error), to
        be able to statistical significance when comparing the the previous baselines
      -  As a qualitative comparison, it could be useful to show some examples, where the proposed anomaly detector outperforms the
         previous approaches.

**Summary Of Recommendation:**


The problem is interesting and useful. The method is reasonable. The experiments conducted on benchmark are convincing. The few issues pointed out above should be addressed.

---

> ### Author Response · Authors · 2022-08-22
> **Reply to Reviewer t48G**
>
> Thank you very much for your valuable feedback. We address your comments and suggestions below.
>
> “The obtained results can, in principle, be reproduced. Even though key resources (e.g., proofs, code, data) are unavailable.”
> - The proof of Theorem 1 (which provides the guarantees on FPR and FNR) is available in Appendix A provided in the supplementary material. We will make the code open source if the paper is accepted for publication. The data used in this paper comes from nuScenes and nuPlan, both of which are already publicly available.
>
> “The sampling is based solely on likelihood which may only produce samples that correspond to the major modes of the data distribution. How can you guarantee the diversity of the set of samples?”
> - To detect when a tail event occurs, we need samples that are representative of the predicted cost distribution. This is achieved by independent and identically distributed (i.i.d.) sampling. Sampling in a non-i.i.d. manner to artificially enhance diversity can be detrimental to the detection of tail events. For instance, if we sample deliberately from the tail while constructing the predicted cost distribution, an otherwise low-likelihood event may not be recognized as a prediction failure because such tail events are already well-represented in the drawn samples. Furthermore, we need i.i.d. sampling to provide guarantees on the FPR and FNR of our detector.
>
> “For the results in Tab.3 it would be useful to provide a measure of statistical dispersion (e.g. standard deviation or standard error), to be able to statistical significance when comparing the the previous baselines”
> - We agree, and have included the standard deviation in Table 3 in the updated article (we provide an updated paper in the comment titled “Updated Manuscript”). We note that the standard deviation is low for all the methods.
>
> “As a qualitative comparison, it could be useful to show some examples, where the proposed anomaly detector outperforms the previous approaches.”
> - In figure 4 and L289-L301, we show and explain examples of our detector and other detectors including examples where our detector outperforms others. Additionally, we have included more qualitative examples from the nuPlan dataset in Appendix B.5.
>
> Overall, we reiterate that our detector improves the safety of the AV stack by identifying potentially safety-critical prediction errors. We also want to emphasize that in addition to the theoretical and experimental contributions, since our method is computationally lightweight, modular, and prediction model agnostic, it is easy to integrate in any AV stack that leverages multi-agent behavior prediction. Hence, we expect this work to have significant adoption in AV stacks.

---

> > ### Author Response · Authors · 2022-08-26
> > **Follow-up Response**
> >
> > Dear Reviewer,
> >
> > Thank you very much for your valuable feedback! Kindly let us know if we have addressed all your concerns. We would be happy to discuss any unresolved points.

---

### Author Response · Authors · 2022-08-22
**Updated Manuscript**

We have responded to the comments provided by the reviewers and incorporated their feedback in an updated draft of the paper attached here. This includes more images for a qualitative analysis of our detector, inclusion of the standard deviation in the results in Table 3, further discussion in certain areas, and edits to further improve clarity. Changes are in blue and we provide the paper and appendix as a single pdf.

---

### Meta-Review · Area_Chair_ZNwf · 2022-08-13

**Recommendation:** Accept (Poster)
**Confidence:** 4

**Metareview:**

The paper addresses an essential problem with a convincing, reasonable approach and analyzes its theoretical guarantees. It is well-written, however, the experimental evaluation needs to be more thorough.

Main pros:
- The paper addresses an essential problem with a convincing approach and analyzes its theoretical guarantees
- The paper is well-written

Main cons:
- There are several concrete technical concerns that were raised by the reviewers and need to be addressed
- More thorough evaluation is needed

Post-rebuttal update: The revised manuscript successfully addressed the main issues.

**Best Paper Nomination:**

No

---

> ### Author Response · Authors · 2022-08-22
> **Reply to Area Chair ZNwf**
>
> We appreciate the time the reviewers spent on the evaluation of our work. We have responded to the comments provided by the reviewers and incorporated their feedback in an updated draft of the paper (we provide an updated paper in the comment titled “Updated Manuscript”). This includes more images for a qualitative analysis of our detector, inclusion of the standard deviation in the results in Table 3, further discussion in certain areas, and edits to further improve clarity. Changes are in blue.
>
> “There are several concrete technical concerns that were raised by the reviewers and need to be addressed”
> - We have clarified the technical concerns raised by reviewers in our responses.
>
> “More thorough evaluation is needed”
> - We have included additional experimental runs for the results in Table 3 and incorporated more qualitative results and their discussion in the appendix of the paper.